# Visuomotor Tracking Task for Enhancing Activity in Motor Areas of Stroke Patients

**DOI:** 10.3390/brainsci12081063

**Published:** 2022-08-10

**Authors:** Toshiaki Wasaka, Kohei Ando, Masakazu Nomura, Kazuya Toshima, Tsukasa Tamaru, Yoshifumi Morita

**Affiliations:** 1Department of Electrical and Mechanical Engineering, Nagoya Institute of Technology, Nagoya 466-8555, Japan; 2Center of Biomedical Physics and Information Technology, Nagoya Institute of Technology, Nagoya 466-8555, Japan; 3Kaikokai Rehabilitation Hospital, Kanie 490-1405, Japan

**Keywords:** neural plasticity, movement-related cortical potentials, grip force, primary motor cortex, visuomotor tracking task, stroke

## Abstract

Recovery of motor function following stroke requires interventions to enhance ipsilesional cortical activity. To improve finger motor function following stroke, we developed a movement task with visuomotor feedback and measured changes in motor cortex activity by electroencephalography. Stroke patients performed two types of movement task on separate days using the paretic fingers: a visuomotor tracking task requiring the patient to match a target muscle force pattern with ongoing feedback and a simple finger flexion/extension task without feedback. Movement-related cortical potentials (MRCPs) were recorded before and after the two motor interventions. The amplitudes of MRCPs measured from the ipsilesional hemisphere were significantly enhanced after the visuomotor tracking task but were unchanged by the simple manual movement task. Increased MRCP amplitude preceding movement onset revealed that the control of manual movement using visual feedback acted on the preparatory stage from motor planning to execution. A visuomotor tracking task can enhance motor cortex activity following a brief motor intervention, suggesting efficient induction of use-dependent cortical plasticity in stroke.

## 1. Introduction

One of the primary goals of rehabilitation following stroke is recovery of motor function. For this purpose, it is necessary to improve muscle strength and motor control processes in both the central and peripheral nervous systems. In stroke patients, the excitability of cortical motor neurons is reduced on the lesion side compared to the contralateral (intact) hemisphere. Normally, the bilateral hemispheres exert reciprocal intercortical inhibition via the corpus callosum, but this becomes asymmetric after stroke, leading to facilitation of contralesional cortical activity and diminished ipsilesional cortical activity. Therefore, restoration of this interhemispheric balance is essential for improving motor function on the affected side, and several noninvasive interventions have been developed to enhance ipsilesional motor output. For example, low-frequency repetitive transcranial magnetic stimulation (rTMS) over the motor cortex of the unaffected hemisphere can enhance excitability of the injured hemisphere by suppressing interhemispheric inhibition [1,2,3,4], whereas high-frequency rTMS can directly enhance neural activity on the ipsilesional side [2].

For sustained improvement of motor control following stroke, it is necessary to induce use-dependent neuroplasticity. If a limb with motor paralysis is not used, the cortical representation is reduced due to lack of synaptic input and concomitant neurotrophic support. For improved motor function of the paretic hand, Taub and colleagues developed constraint-induced (CI) therapy in which use of the unaffected side (controlled by the contralesional hemisphere) is constrained to promote increased use of the affected side [5,6]. Short term (12-day) CI therapy is meant to induce use-dependent brain plasticity and promote cortical reorganization in the lesioned hemisphere, and a recent neuroimaging study indeed reported that the cortical representation of the paretic hand was enlarged and motor performance greatly improved by the end of the intervention [7]. However, it is unclear whether a brief motor intervention for the paralyzed limb can induce neuroplasticity and functional recovery. Therefore, the objective of the present study was to examine if a feedback-based motor control task can enhance ipsilesional motor cortex activity after only a relatively small number of repetitions.

Feedback, especially somatosensory information, is needed to improve motor performance through motor learning [8]. However, motor disorders are often accompanied by sensory disorders as many nerves transmit both sensory and motor signals, and adjacent sensory and motor cortices are frequently within the lesion zone. Thus, the state of motor exertion may not be perceived by somatosensory feedback. A previous study also demonstrated that presenting force exertion as visual feedback can improve force control of the paretic hand, suggesting that visual information can facilitate motor learning [9]. Therefore, the present study adopted a visuomotor tracking task, allowing subjects to control the state of muscle force using visual information, and tested the hypothesis that this motor task can enhance motor cortex activity and control of the paretic hand compared to a task without feedback.

The present study investigated changes in motor cortex activation immediately before and after a motor task involving the paretic limb. Movement-related cortical potentials (MRCPs) can be measured prior to the onset of self-initiated muscle contraction [10] from the motor areas innervating the contracting muscle groups [11,12], and a comparison of MRCPs before and after motor intervention revealed changes in motor activity during all stages from preparation and planning to motor execution. In addition, since MRCP amplitudes change concomitantly with recovery of motor function in stroke [13], these changes can be used as a verification of rehabilitation efficacy.

## 2. Materials and Methods

### 2.1. Participants

A total of 21 patients (14 males and 7 females, mean age 61.3 ± 11.1 years old) currently receiving rehabilitation following stroke (mean 66.8 days poststroke) at Kaikokai Rehabilitation Hospital and 6 healthy right-handed volunteers (3 males and 3 females, mean age 45.3 ± 6.0 years old) participated in this study. The inclusion criteria were as follows: (1) experiencing a stroke for the first time; (2) sufficient cognitive ability to understand and appropriately perform a motor task; (3) ability to perform tasks with paralyzed hands; and (4) no visual impairment and field defect. The exclusion criteria were as follows: (1) other neurological problems or orthopedic injuries; (2) aphasia that would make intervention difficult; and (3) recent participation in other rehabilitation research or drug experiments. The clinical conditions of stroke patients are summarized in Table 1. The study was approved by the ethics committee of Kaikokai Rehabilitation Hospital: 2018-1, Nagoya Institute of Technology: 2021-16, and all participants provided written informed consent prior to measurements.

### 2.2. Two Manual Tasks of Motor Intervention

The visuomotor tracking task (VM task) was conducted using a device that quantitatively evaluates adjustment of grip force (iWakka, Aimu Co., Moriyama, Japan). Participants were requested to accurately match the target hand grip force presented on a computer monitor using the paretic hand, with visual feedback showing the force exerted provided on the same monitor [14]. Briefly, the participant fixated on a point at the center of the monitor that moved up and down depending on the amount of force exertion. The target force was presented in advance and during the task. The target force line was set as a “mountain” shape and ranged between a minimum value of 150 g and maximum value of 400 g. At the beginning of the VM task, the participant adjusted the grip force to 150 g. The target force value increased by 62.5 g over 4 s at a constant speed and remained at this new level (212.5 g) for 4 s before rising again by 62.5 g to a new plateau (275 g). This sequence was repeated two additional times until the maximum value was reached. The reverse sequence was then presented until the target force reached 150 g. In the control task (C task), the participants repetitively increased grip force from 0 to 400 g at their own pace. In the control task, no objective visual feedback was provided as the monitor was covered, but participants were instructed to perform the task while looking at the hand. The duration of one trial was 90 s for both tasks, and the participants repeated both tasks twice without a break.

### 2.3. Experimental Design

Figure 1 illustrates the experimental procedure and time line of MRCP measurements. A participant was seated on a chair in the experimental room during all tests. Patients completed VM and C tasks using the paretic hand, while healthy controls used their left hand. Each motor intervention task was conducted on a separate day with an interval of at least three days between tests. The order of performance was randomized, with half of the participants completing the VM task first and half completing the C task first.

### 2.4. Electroencephalogram Recordings

Electroencephalograms (EEGs) were recorded from scalp positions Fz, Cz, Pz, C3, and C4 according to the international 10–20 system with linked earlobes as the reference (using a Polymate2 AP216, Miyuki Giken, Tokyo, Japan). Impedance at all EEG electrodes was less than 5 kΩ. The EEG signals were bandpass-filtered at 0.15–100 Hz, and both EEG and grip force measures were stored on a personal computer at a sampling rate of 500 Hz. MRCPs were recorded during hand grips to a target force of 400 g repeated every 5–7 s at the participant’s own pace. Participants performed 100 of these grip movements (5 sets of 20) before the motor intervention (VM or C) and another 100 grip movements (also in 5 sets of 20) after the intervention.

In off-line analysis, we obtained MRCP waveforms using the onset of gripping force as a trigger. Force onset was defined as the instance when force exceeded a 5% level of the target. The analysis period was from 2500 ms before to 1000 ms after movement onset. Electrophysiological signals were then low-pass filtered at a 20 Hz cutoff. For signal averaging, the trials containing EEG deflections exceeding 100 µV were excluded because it was assumed that these arose from unintended movements or blinks, and were thus contaminated. The data for the first 500 ms (−2500 to −2000 ms before movement onset) were used to calculate the baseline MRCP amplitude. For calculation of mean preparatory MRCP amplitude, the preparatory period for grip force was divided into two subperiods preceding force onset, from −2000 to −500 ms and −500 to 0 ms.

All patients completed 200 MRCP measurement trials and both motor intervention tasks. However, the EEG data of five patients and one normal subject were excluded from the analysis due to numerous artifacts from blinks and unintended muscle contractions during the preparatory period of grip movements. Therefore, we analyzed the MRCP data from 16 patients and the 5 healthy control participants.

### 2.5. Statistical Analyses

We separately investigated how motor intervention affected MRCP amplitude in the damaged and intact hemispheres because we hypothesized that effect of the visuomotor tracking task influenced the injured hemisphere. Mean MRCP amplitudes measured from ipsilesional and contralesional electrodes (C3 or C4 depending on the lesion side) were compared using Friedman non-parametric tests on pre- and post-intervention of the VM and C tasks. Since all normal subjects used the left hand, signals were measured from C4 (the electrode covering the hand area of the right motor cortex). Post hoc analyses were conducted using the Wilcoxon signed-rank test with Bonferroni correction. A *p*-value of 0.05 was considered significant for all tests.

## 3. Results

### 3.1. Comparison of Gripping Force during MRCPs Measurement

The mean peak amplitudes and latencies of grip force during MRCP measurements are presented in Table 2. In the statistical analysis using the Friedman test, there were no significant differences in force values between pre- and post-MRCP recording periods for either intervention. Since the forces exerted by the paretic hand were similar during both tasks, it is probable that the gripping movements were also similar.

### 3.2. MRCP Measures from the Ipsilesional Hemisphere

Figure 2 shows the grand-averaged MRCP waveforms obtained from the ipsilesional hemisphere (electrode C3 or C4 of the 10–20 system depending on lesion side) before (Pre) and after (Post) the VM and C intervention tasks. The MRCP waveforms recorded during the preintervention (Pre-C) measurement session at 1000 ms preceding force onset did not differ from those recorded at the equivalent time point during the postintervention measurement session (Post-C). In contrast, the corresponding waveforms recorded after the VM task (Post-VM) were larger in amplitude than those recorded before the intervention (Pre-VM). Figure 3 shows the mean MRCP amplitudes occurring within the −2000 to −500 ms and −500 to 0 ms subperiods relative to force onset before and after the interventions. The results revealed a significant difference in MRCP amplitude within the −500 to 0 ms subperiod (*p* < 0.01), while the post hoc test showed that the mean amplitude was significantly higher Post-VM than Pre-VM (*p* < 0.01). In contrast, amplitude did not differ significantly for Post-C compared to Pre-C. Moreover, MRCP amplitude was not significant within the −2000 to −500 ms subperiod.

### 3.3. MRCP Measures from the Contralesional Hemisphere

These same statistical analyses revealed a significant result on MRCPs within the −2000 to −500 ms subperiod recorded from the uninjured hemisphere (*p* < 0.05). Post hoc tests showed no significant differences among pre- and post-intervention of two motor tasks.

### 3.4. MRCP in the Normal Subjects

There was no significant change in the two subperiods among pre- and post-intervention of the two motor tasks.

## 4. Discussion

A brief visuomotor tracking task requiring grip movement of the paretic hand with visual feedback of force generation enhanced neural activity in the ipsilesional motor area of stroke patients as measured by MRCP amplitude, thereby achieving a primary goal of neurorehabilitation. In contrast, the same duration of grip movement without visual feedback of force exertion had no effect on ipsilesional MRCP amplitude. Although the sample size of normal subjects was small, there was no change in MRCP amplitude with the motor intervention. Therefore, this visuomotor tracking movement task may be a promising intervention for facilitating motor recovery following stroke.

Previous studies have reported a positive correlation between MRCP amplitude and muscle force generation [15,16,17] as greater muscle force requires the requirement of larger numbers of motor neurons. However, the target force level was the same during pre- and post-intervention MRCP recordings, and average force exerted did not differ between pre- and post-intervention sessions. Therefore, greater force generation cannot account for the observed increase in MRCP amplitude. Another factor that modulates MRCP amplitude is fatigue [18,19,20]. Although the target force was not high, performing the MRCP measurements with the paralyzed hand may cause fatigue. However, since the number of MRCP measurement trials was the same before and after both interventions, it is unlikely that fatigue can account for the MRCP enlargement following the VM intervention. Rather, further analysis suggested that MRCPs increased in magnitude due to greater recruitment and activity of preparatory motor neurons.

Stroke patients exhibited a significant enhancement in MRCP amplitude after the VM task during the 500 ms subperiod just before movement, whereas healthy control subjects showed no change. The MRCPs preceding voluntary movement reflect the activation of cortical motor neurons involved in movement planning and preparation. This is reflected by a slow negative potential starting about 2 s prior to self-initiated voluntary movement [10]. Based on topographical distribution and the kinetics of amplitude development, MRCPs are divided into two components: the Bereitschaftspotential (BP) and the negative slope (NS). The BP begins about 2 s before movement onset, whereas the NS starts about 500 ms prior to movement in the central region contralateral to the contracting muscle groups [21]. We assume that the BP and NS components appear in the −2000 to −500 ms and −500 to 0 ms periods, respectively. During the NS component of preparatory MRCPs, subdural records revealed that activity arose mainly from the contralateral primary motor cortex (MI) and the bilateral supplementary motor area (SMA) [11,12]. Therefore, our grip force adjustment task with visual feedback likely enhanced neural activities in the SMA and M1 of stroke patients.

Motor recovery from stroke requires the induction of neuroplastic changes in the damaged hemisphere. For instance, transcranial direct current stimulation, which is believed to induce such neuroplasticity, can promote the recovery of skilled motor function [22]. In addition to these noninvasive brain stimulation interventions, plastic changes induced by repeated motor practice are also crucial for motor recovery [23,24]. We found that only two sets of the 90 s VM task significantly increased ipsilesional MRCP amplitudes, whereas no such changes were achieved using a simple gripping task without visuomotor feedback. Both motor interventions activated the motor areas innervating the paretic fingers as evidenced by MRCP measurements, suggesting that visual feedback of the force exerted during the VM task but not the visual feedback of hand movement during the C task promoted recruitment of preparatory motor neurons for controlling movement. Imaging studies have reported that controlling the grip force during a VM task demands integration of motor output and sensory information in the brain, a process involving circuits spanning frontal, sensorimotor, and parietal cortices [25,26,27]. Further, both somatosensory and visual information are required for motor planning and control. However, the paralyzed limb does not provide somatosensory information to the cortex, so visual information is more important. In stroke patients with paresis of the upper limb, this VM task may enhance cortical processing that facilitates relearning of impaired motor skills by promoting continuous integration of motor signals for force control using visual signals conveying force exertion.

We found that a single training session of this VM task enhanced activation of motor neurons involved in motor preparation. These results indicate that the use-dependent plasticity mediating motor learning requires sensory feedback on force generation, rather than simply movement repetition. There are two stages of motor learning involving distinct patterns of neural activity [28]. The second learning process requires changes in cortical representations induced during long-term motor practice [29,30]. Therefore, it is the first learning process that likely contributed to MRCP amplitude modulation following the VM intervention.

Electroencephalographic studies have shown that movement on the paralyzed side is accompanied by greater cortical activity in the contralesional hemisphere due to disinhibition from reduced inhibitory output from the ipsilesional hemisphere [31,32], resulting in imbalanced activity and insufficient motor output. By applying 1 Hz rTMS to the contralesional MI, this transhemispheric inhibition can be suppressed, thereby facilitating motor output from the damaged hemisphere [33]. However, this VM intervention did not alter neural activity on the contralesional side as MRCP amplitudes were unchanged. We therefore speculate that the VM task acts directly on motor areas in the ipsilesional hemisphere.

Rehabilitation during the acute period following stroke is essential for long-term recovery of function, and a critical focus during this period is enhancing corticospinal excitability [34,35]. Our study was conducted about 60 days following stroke and the results showed MRCP enhancement in the central region of the contralateral (ipsilesional) hemisphere, suggesting elevated activity in motor, premotor, and supplementary motor cortices. Collectively, these results suggest that it is possible to promote use-dependent plasticity of relevant motor areas (i.e., those representing a paretic hand) using a brief motor intervention with visuomotor feedback.

Our study has certain limitations. First, the long-term retention of the effect of the VM is unclear. Since obtaining MRCPs takes approximately 12 min, the VM task has been to shown to enhance the activities in the motor areas in the ipsilesional hemisphere during this period. Second, the motor intervention effects varied between patients with stroke, and the detailed effect of motor intervention still remains unclear. We investigated the correlation between the score of Fugl–Meyer Assessment and MRCP amplitude, but observed no significant relationship. By clarifying the relationship between the degree of brain damage, paralysis, and plastic change in brain activity after motor intervention, its application to rehabilitation is expected to be promoted. Third, the motor intervention induced MRCPs amplitude enhancement in stroke patients in the recovery stage, but it is unclear whether it would be effective in other stages of stroke. It is worth examining the effect of motor intervention on the acute or chronic stroke phase in the future study.

## 5. Conclusions

We demonstrate that a brief visuomotor tracking task using the paralyzed hand can increase the activities in motor-related areas of the lesioned hemisphere in stroke patients, thereby achieving a primary goal of neurorehabilitation. This improvement relies on compensation for lost somatosensory feedback using visual feedback of force generation, and ultimately results in neuroplastic changes that facilitate the recruitment of preparatory motor neurons in M1, the premotor cortex, and the supplementary motor cortex. However, further studies are needed to clarify the molecular mechanisms mediating these neuroplastic changes and the duration of these effects.

## Figures and Tables

**Figure 1 brainsci-12-01063-f001:**
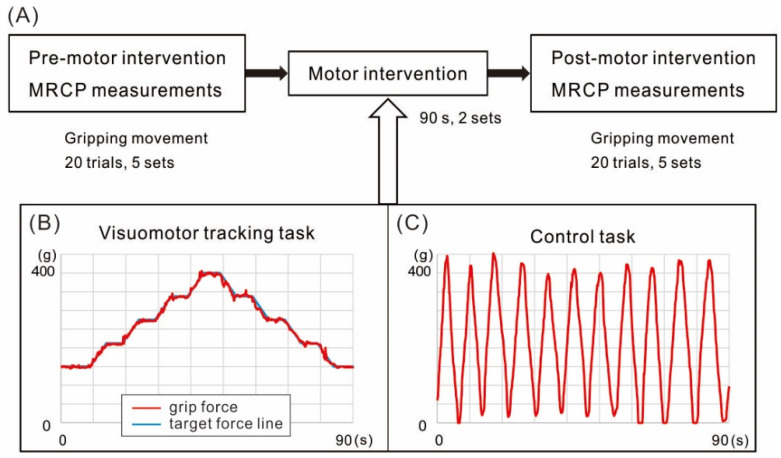
Schematics of the experimental design and motor interventions. (**A**) The experimental design. Movement-related cortical potentials (MRCPs) were measured from the scalp over motor cortex before (Pre) and after (Post) the two motor interventions. Participants performed hand grip movements every 5–7 s to a peak target force of 400 g. One measurement session was comprised of five sets of 20 trials, with one-minute breaks between sets. Patients preformed these movements with the paretic hand and healthy controls with the left hand. (**B**) The visuomotor tracking task (VM) required participants to replicate the hand grip force pattern shown on a computer screen with simultaneous visuomotor feedback. (**C**) The control task (**C**) required participants to perform repetitive hand grips to 400 g at their own pace with visual observation of the hand but no visuomotor feedback.

**Figure 2 brainsci-12-01063-f002:**
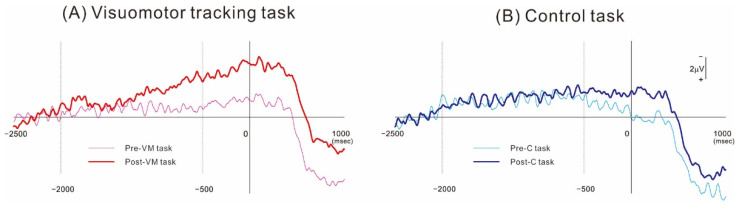
Grand-averaged MRCP waveforms obtained from over the ipsilesional motor cortex (C3 or C4 position of the 10–20 system). (**A**) MRCPs obtained before and after the visuomotor tracking task of motor intervention. (**B**) Corresponding MRCPs obtained before and after the control task. In both MRCP measurement conditions, the voltage deflection started about 2 s before movement onset. The MRCP amplitude was greater after the visuomotor tracking task but not after the control task.

**Figure 3 brainsci-12-01063-f003:**
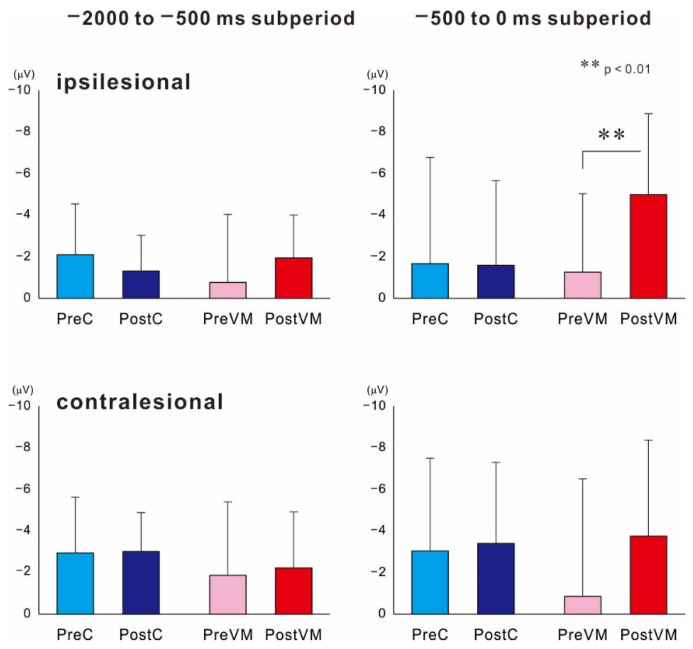
Mean amplitudes of MRCPs recorded within −2000 to −500 ms and −500 to 0 ms subperiods from ipsilesional and contralesional motor cortices before (Pre) and after (Post) the motor intervention (C or VM). In the damaged hemisphere, mean MRCP amplitude after the VM task (Post-VM) was significantly enhanced during the −500 to 0 ms subperiod but not during the −2000 to −500 ms subperiod. There were no changes in MRCP amplitudes during either subperiod after the control intervention. ** *p* < 0.01, statistical significance compared within two pairs.

**Table 1 brainsci-12-01063-t001:** Clinical characteristic of stroke patients.

Patient	Age (Years)	Sex	Time since Stroke (Days)	Handedness	Type of Stroke	Paretic Side	FMA	MMSE
1	69	M	59	Right	infarction	Left	27	27
2	67	F	38	Right	infarction	Right	58	30
3	65	M	145	Right	hemorrhage	Right	63	14
4	70	F	100	Right	hemorrhage	Right	54	23
5	83	M	72	Right	infarction	Right	60	24
6	62	M	91	Left	infarction	Right	60	26
7	49	M	43	Right	infarction	Right	65	28
8	57	M	51	Right	infarction	Right	61	30
9	45	M	54	Right	infarction	Left	62	30
10	59	M	48	Left	hemorrhage	Right	61	29
11	62	F	92	Right	hemorrhage	Left	57	25
12	53	M	72	Right	hemorrhage	Right	60	30
13	71	M	120	Right	infarction	Right	64	21
14	48	F	24	Right	hemorrhage	Right	59	30
15	54	M	63	Right	hemorrhage	Right	50	30
16	68	F	73	Right	hemorrhage	Right	45	18
17	67	F	92	Right	hemorrhage	Right	59	30
18	36	F	64	Right	infarction	Left	18	30
19	75	M	29	Right	infarction	Right	62	26
20	59	M	37	Right	infarction	Right	59	30
21	68	M	36	Right	infarction	Right	33	26

M = male; F = female; FMA = Fugl–Meyer Assessment; MMSE = Mini-Mental State Examination.

**Table 2 brainsci-12-01063-t002:** Peak amplitudes and peak latencies of grip force before the control and visuomotor tracking tasks.

**Peak** **Amplitude (g)**	**Control Task**		**Visuomotor Tracking Task**	
	**Pre**	**Post**	**Pre**	**Post**
Mean	411.7	404.6	406.5	407.2
SD	44.1	41.8	36.5	39.8
**Peak** **Latency (ms)**	**Control Task**		**Visuomotor Tracking Task**	
	**Pre**	**Post**	**Pre**	**Post**
Mean	345.3	333.6	320.8	306.8
SD	161.2	153.2	164.5	160.5

## Data Availability

The data available from the corresponding author upon reasonable request.

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
