# Peer review of "Visuomotor Tracking Task for Enhancing Activity in Motor Areas of Stroke Patients"

_brainsci, 2022, doi:10.3390/brainsci12081063_

Round 1
Reviewer 1 Report
Wasaka and colleagues looked at motor training-induced changes in MRCP post-stroke. They showed that their visuomotor tracking task enhanced MRCP amplitudes and thus motor activity in the ipsilesional hemisphere. The paper is well written and the study is generally well done. I have a couple of small comments about the stats and results, outlined below.
The control group is very small and I don’t see any results reported for this group. What was the purpose of including them if not to compare with the stroke group? If they are going to be included, I think it would be important to do a power calculation and adjust the sample size accordingly.
The authors state that they ran a 2-way RM ANOVA with the factors of time and intervention, which is in line with the purpose of the study. Was this done separately for the ipsilesional and contralesional cortices? Weren’t the participants performing the task with their paretic hand? Therefore, I’m not sure why the contralesional data is included. I’m not sure what it adds if not directly compared to the ipsilesional side. Please reference the importance of this.
In the results, I would suggest starting with the results from the statistics and then going into more detail. This would improve clarity and reduce repetition.
Author Response
We would like to thank the Reviewer for the comments and suggestions. To conduct the experiment, we assumed that healthy subjects showed no change in brain activity with 400 g of grip force movement both in the C and VM task, and verified it. Therefore, we added the MRCP amplitude for normal subjects in the Results.
In the ipsilesional and contralesional hemisphere, we revised the description of the statistical analysis. We analyzed MRCP in both hemispheres separately as we hypothesized that the motor intervention effect mainly influenced the injured hemisphere. Moreover, we moved the description of MRCP analysis from the beginning of the Results to the Methods section.
Reviewer 2 Report
“Visuomotor tracking task for enhancing activity in motor areas of stroke patients”
Overall strengths of the article:
This paper investigated changes in motor cortex activation immediately before and after a motor task involving the paretic limb post-stroke. Movement-related cortical potentials (MRCPs) are measured prior to the onset of self-initiated muscle contraction from the motor areas innervating the contracting muscle groups. The objective was to examine if a feedback-based motor control task can enhance ipsilesional motor cortex activity after only a relatively small number of repetitions. MRCP amplitudes change concomitantly with the recovery of motor function in stroke, these results suggest that it is possible to promote use-dependent plasticity of relevant motor areas using a brief motor intervention with visuomotor feedback. This is a very interesting manuscript. I have minor concerns that should be addressed before publication.
Specific comments on weaknesses:
1. This study was conducted 60 days following stroke; if the intervention was used in chronic stroke patients (e.g., more than 1 year after strokes) how the results have been changed?
2. Number of subjects is very low, keeping in mind that some of them were removed from the final analysis.
3. Line; 59-60 Typo; ‘However, motor disorders are often accompanied by sensory disorders as many nerves transmit both sensory and ‘more’ signals, and adjacent sensory and more cortices are frequently within the lesion zone”. I think they mean ‘motor’ signals.
Author Response
- We would like to thank the Reviewer for the comments. We recruited convalescent patients of recovery stage in the hospital. Since it has been reported that rehabilitation using long-term motor intervention in the chronic phase of stroke improves motor function, it is expected that the motor intervention in this study should be increased the brain activity in the motor areas. It is worth examining the effect of motor intervention on the acute or chronic stroke phase in the future study. In this point, we added the description in the Discussion section.
-
The main purpose of present study was to clarify effective motor intervention that enhance the activity of the motor areas of stroke patients, but to compare the motor intervention effect between stroke and healthy subjects. To conduct the experiment, we assumed that healthy subjects showed no change in brain activity with 400 g of grip force movement both in the C and VM task, and verified it. We agreed with your comments, and we could not strongly mention the motor intervention effect to the normal subjects. We added the description that motor intervention was useful for stroke with impairment of motor function in the Discussion section.
- We apologized the typographical error. We revised the mistake. We have corrected it.
Reviewer 3 Report
Wasaka et al.'s paper on visuomotor tracking task for enhancing activity in motor areas has limited novelty in the study. There are several similar or better paper exist on the same topic with more task to prove the hypothesis. So, in my opinion the paper would be rejected from this journal.
Author Response
We would like to thank the Reviewer for the comments. Our study investigated the plastic changes in the brain activity caused by a single motor intervention for stroke. The results showed the possibility of effective motor intervention that increased neural activity in the motor areas. We believe that our results contribute to stroke rehabilitation.
Reviewer 4 Report
Thank you for the opportunity to review this valuable work. I would like to discuss some issues that may need to be addressed in the manuscript.
Materials and Methods
Participants
- Describe inclusion and exclusion criteria when recruiting study subjects.
Results
Table1.
- In patient 11, the MMSE score is 45. As far as I know, the MMSE is the highest score of 30. Which MMSE version did you use?
Discussion
Please describe the limitations of this study.
Author Response
- Describe inclusion and exclusion criteria when recruiting study subjects.
We thank the reviewer for their comments. We have added the inclusion and exclusion criteria in the Methods section.
- In patient 11, the MMSE score is 45. As far as I know, the MMSE is the highest score of 30. Which MMSE version did you use?
We apologized as we mistakenly wrote an incorrect value of the MMSE score of patient 11. We have corrected it to 25.
- Please describe the limitations of this study.
Our results showed the modulation in motor areas for motor function recovery for stroke, but tour results have some limitations, which we included it in the Discussion.
Reviewer 5 Report
The research question investigated in the present manuscript is intriguing and timely. While I believe the experimental method is off interest to readers, the statistical analyses performed were not sufficiently justified. Importantly, it is unlikely that the data collected wer normally distributed, and thus non-parametric tests/resampling statistics should be used. Additionally, how are the different factors that vary across individual patients taken care of during the statistical analyses or interpretation of the results. Given that the proposed clinical applications of this work, I suggest that the authors extend their analyses to take across-subject variability into account. Finally, it will be helpful if the authors could clearly illustrate significance level in all of the result figures.
Author Response
We would like to thank the Reviewer for the comments. As pointed out, since our data is not normally distributed (using the Shapiro-Wilk normality test), we adopted non-parametric analyses (Friedman test and Wilcoxon signed-rank test). The results showed a significant MRCP amplitude enhancement after the VM task that was the same as the results before the correction. We have subsequently revised the description in the Methods and Results.
We agreed with the comment of Reviewer concerning the individual variability and tried to analyze the correlation between the FMA score and MRCP amplitude. The results showed no significant relationship between them. We think that the number of patients in this study is insufficient to clarify the relationship between the degree of brain damage and paralysis and brain activity after motor intervention. We have included this point in the limitation of our study in the Discussion section.
Significant MRCP amplitude enhancement could only be observed in the 0 to -500 ms subperiod after the VM task. To emphasize this result, we have made minor revisions to Fig. 3.
Round 2
Reviewer 3 Report
Looking good. Go ahead with acceptance of paper.